# Optimized tDR Sequencing Reveals Diversity and Heterogeneity in tRNA-Derived Fragment Landscapes in Mouse Tissues

**DOI:** 10.3390/ijms26188772

**Published:** 2025-09-09

**Authors:** Daisuke Ando, Sherif Rashad, Kuniyasu Niizuma

**Affiliations:** 1Department of Neurosurgical Engineering, Graduate School of Biomedical Engineering, Tohoku University, Sendai 980-8575, Japan; 2Division of Development and Discovery of Interventional Therapy, Tohoku University Hospital, Sendai 980-8574, Japan; 3Department of Translational Neuroscience, Tohoku University Graduate School of Medicine, Sendai 980-8575, Japan; 4Department of Neurology, Tohoku University Graduate School of Medicine, Sendai 980-8574, Japan; 5Department of Neurosurgery, Tohoku University Graduate School of Medicine, Sendai 980-8574, Japan

**Keywords:** tRNA fragments, G4 quadruplexes, tDRs, translation, angiogenin, epitranscriptome, tRNA sequencing

## Abstract

Transfer RNA-derived small RNAs (tDRs) are increasingly being recognized as versatile regulators, yet their physiological landscape remains poorly charted. We analyzed tDR expression in seven adult mouse tissues to explore tissue-specific tDR enrichment using a tDR-optimized library preparation methodology. We catalogued 26,901 unique nuclear tDRs (ntDRs) and 5114 mitochondrial tDRs (mtDRs). Clustering analysis segregated the tissues, with the spleen and lungs forming a distinct immune cluster. Tissue-versus-all and pairwise differential analysis showed the spleen harboring unique ntDRs and mtDRs. Tissue-enriched tDRs arose from specific isoacceptor and isodecoder tRNAs, independent of mature tRNA abundance, suggesting selective biogenesis rather than bulk turnover. G-quadruplex prediction revealed a pronounced enrichment of potentially quadruplex-forming ntDRs in the kidneys, heart, and spleen, predominantly derived from i-tRFs and tRF3 fragments, suggesting structure-dependent functions in immune regulation. We also benchmarked our library strategy against the PANDORA-seq method. Despite comparable or lower sequencing depth, our method detected ~3–10-fold more unique ntDRs and we observed a clearer representation of tRF-3 fragments and greater isotype diversity. Our tissue atlas and improved tDR sequencing method reveal extensive tissue-specific heterogeneity in tDR biogenesis, sequencing, and structure, providing a framework for understanding the context-dependent regulatory roles of tDRs.

## 1. Background

tRNA-derived small non-coding RNAs (tDRs) are central to a rapidly expanding research field [1,2]. Initially discovered as stress-induced angiogenin-mediated cleavage products of tRNAs, yielding five prime and three prime tRNA halves (or 5′tiRNAs and 3′tiRNAs (tRNA-derived stress-induced RNAs)) [3,4,5], tDRs have grown in complexity, with many subclasses identified as a result of various modes of tRNA processing [6,7]. In addition to angiogenin, several other RNAses have been reported to generate tDRs [6,8]. These enzymes act on specific tRNAs in specific contexts [9,10,11,12]. tRNA modifications also play important roles in regulating tDRs biogenesis and function [13,14,15,16]. Thus, the production of tDRs is currently understood to be a highly ordered and regulated process and not as a simple degradation byproduct of tRNAs [1,5]. Nonetheless, our understanding of tDRs biogenesis remains incomplete, especially in physiological conditions.

Most research performed on tDRs focused on their roles in cellular stress or diseases such as cancer, neurodegeneration, and metabolic disorders [2,17,18,19,20,21,22]. Thus, how tDRs shape gene expression and the proteome in physiological states remains understudied. Several studies have shown how tDRs regulate stem cell function and differentiation capacity [23,24] and β-cell maturation [25]. Mechanistically, tDRs interact with RNA-binding proteins (RBPs) to displace them from target mRNAs [26,27]. At the level of translation, tDRs can stimulate ribosomal mRNAs translation [28,29]. On the other hand, stress-induced 5′tiRNAs suppress translation initiation [3]. At the transcriptional level, tDRs were proposed to act as miRNA-like molecules to induce gene silencing [30]; however, this remains debated because, unlike microRNAs, tDRs can form secondary structures and G4 quadruplexes that can impact their function [1,17,31]. 5′tiRNAs can promote corresponding tRNA gene transcription [32], while tRF3s (18~22 nt fragments originating from the 3′ side of the tRNA) regulate LTR-retrotransposon [33]. Thus, tDRs act through diverse mechanisms rather than a single mode of activity. This heterogeneity in their function adds a layer of complexity to tDRs. tDR modifications, sequences, origins, sites of production, secondary structures, and other factors likely determine the diverse ways in which tDRs act [1,26,31,34,35]. Given the clear links between tDRs and diseases, either as potential therapeutic targets for cancers and other diseases or as biomarkers for various conditions [36,37,38], understanding their biogenesis and function is critical. However, no single approach is sufficient; combinations of biochemical, molecular, and computational strategies are necessary. Importantly, more high-quality datasets cataloging tDR expression under physiological and pathological conditions are needed. Unfortunately, the current dependence on less optimal datasets, such as tDR sequences derived from *The Cancer Genome Atlas* (TCGA) [39,40], in which the sequencing technique was not optimized to capture tRNAs nor tDRs (especially considering the currently accepted practices [41,42]), have caused some conflicting conclusions in terms of tDRs biogenesis and function. Thus, newer datasets with better sequencing techniques and deeper coverage of tDRs are needed. One of the important approaches in studying non-coding RNAs is cataloging their expression in different cell states, tissues, and cell types. While this has been performed effectively for microRNAs, for example (which are easier to sequence and have more robust over the shelf tools to prepare the libraries [43,44,45]), such efforts are lacking in the tDR field. While several databases have catalogued tDRs from hundreds of published sequencing datasets [46,47], these databases did not include only datasets optimized for tDR detection; thus, biases in tDR expression should be expected. Only one study systematically analyzed tDRs expression in different mouse tissues [43]. However, the library preparation for that study also did not include important steps, such as demethylation, deacetylation, and end repair, that are known to increase the coverage of tDRs and avoid biases from modifications and aminoacylation [41,48]. Thus, in this study, we analyzed our previously published tRNA/tDR-optimized small RNA sequencing dataset of seven mouse tissues [49]. We aim to provide an in-depth understanding of the heterogeneity of tDRs across different tissues that could be linked to specific gene regulatory processes and tissue/cell physiology. We further benchmark our method against a previously published dataset to demonstrate how adding certain pretreatment steps to tDR library preparations could drastically increase the depth of tDR detection.

## 2. Results

### 2.1. Global Patterns of tDRs Expression in Seven Mouse Tissues

To explore the tissue-specific expression of tDRs, we used our previously published small RNA-seq dataset of seven different mouse tissues, with three biological replicates per tissue [49]. In this study [49], small RNA fractions (<200 nts) were deacylated, demethylated, and end repaired. These processed allowed for a higher depth of coverage of tDRs to be achieved [41,48], as well as the detection of mature tRNAs [49,50]. However, without such library preparation strategies, tDRs are usually underrepresented in libraries and their detection will be biased due to many factors, such as RNA modifications or aminoacylation [41,48,50,51].

To detect tDRs, we used tDRnamer [8], which not only gives us the naming convention of the detected tDRs [52], but also provides other information that enables in-depth analyses of the following: Sprinzl start/end position, tRNA isoacceptor and isodecoder mapping of tDRs, tDR sequences, etc. We analyzed the data using tDRnamer to detect nuclear-encoded tDRs (ntDRs, i.e., those arising from nuclear-encoded tRNAs) and mitochondrial tDRs (mtDRs) in separate runs. We filtered the detected tDRs to select between 15 and 50 nucleotides in length and have at least 10 counts in any sample for downstream analysis and created a custom classifier to classify the detected tDRs based on their start and end positions on mature tRNAs (Figure 1A,B). Previously, it was shown that the absolute amounts of tRNAs differ between tissues, with spleen and lung being more enriched in other non-coding RNAs, reducing the reads mapped to tRNAs [49]. Analysis of the detected tDR reads revealed similar patterns to those reported previously (Figure 1C,D), with the spleen having the lowest levels of detected ntDRs and mtDRs. The heart had the highest levels of detected mtDRs (more than three-fold in comparison with the next tissue), also following what was reported previously [49]. Overall, we detected 26,901 unique ntDRs and 5114 unique mtDRs (Appendix A).

Examining the subclasses of all detected tDRs, the most abundant in ntDRs were tRF3b, i-tRFs, and 5′tiRNAs (5′ halves or tiRNA_5 in the plot) (Figure 1E). Indeed, variations between tissues were observed. For example, the spleen had the highest relative levels of 5′tiRNAs and tRF5c, while the lung and spleen had the lowest relative levels of i-tRFs. In the mtDRs, the dominant subclasses were i-tRFs and tRF3b, with variable levels between tissues; overall, more i-tRFs were detected in mtDRs relatively compared to ntDRs (Figure 1F). The isotype source of tDRs, i.e., the amino acid isotype of the source tRNA, also showed considerable variability between tissues and between mtDRs and ntDRs (Figure 1G,H). Nonetheless, certain isotype tRNAs represented a considerable source of tDRs such as Glu, Arg, Gly, and His in ntDRs and Glu, Pro, and Thr in mtDRs, with variations between tissues in terms of the relative abundance of tDRs from these sources.

The size distribution analysis revealed greater heterogeneity in ntDR lengths compared to mtDR lengths (Figure 2A,B). In addition, the sparse partial least square regression discriminant analysis (sPLS-DA)-based clustering revealed different clustering patterns between the tissues when using ntDRs or mtDRs (Figure 2C,D). ntDRs analysis by sPLS-DA revealed the unique clustering of the spleen compared to all other tissues, with clustering of the kidneys and lungs together and the remaining tissues together. On the other hand, the brain was the most uniquely clustered tissue on the sPLS-DA analysis of mtDRs. Examining the top 50 variable tDRs, either ntDRs (Figure 2E) or mtDRs (Figure 2F) showed that the spleen and lung tended to cluster together in a distinct cluster compared to other tissues.

In summary, tDRs expression reveals inter-tissue heterogeneity in variable aspects of tDRs such as source tRNAs and biogenesis. tDRs expression could also discriminate between different tissues as shown in the clustering data presented.

### 2.2. tDRs Show Tissue-Specific Enrichment Patterns

To understand the different patterns of tDRs enrichment in tissues, we conducted differential tDR expression analysis, comparing each tissue to all other tissues (for example, brain vs. all, heart vs. all, etc.). To avoid the dilution of mtDRs patterns, we analyzed ntDRs and mtDRs separately. Analysis of differentially expressed ntDRs revealed that the tissue most divergent from others was the spleen (Figure 3A–D, Appendix A). The liver and muscle did not show significantly enriched tDRs when compared to all other tissues. Spearman correlation analysis between different tissues using log2FC values of tDRs revealed modest–no correlation between all tissues, with the strongest correlation being a negative one between the heart and the spleen, the lung, and the kidney (Rho = −0.36 to −0.38) (Figure 3E). The same case was apparent in mtDRs, where the spleen showed the highest number of significant tDRs (Figure 3F–H, Appendix A). However, most tissues did not exhibit significant diversity in their mtDRs expression. Correlation analysis between different tissues using log2FC of mtDRs showed the same weak overall correlation between tissues (Figure 3I).

Because the tissue vs. all approach may introduce bias from high inter-tissue variances, we used EdgeR to conduct a pairwise tissue vs. tissue differential tDRs expression analysis followed by quasi-likelihood post hoc analysis. In both the ntDRs analysis and the mtDRs analysis, the spleen was found to be the most diverse tissue; this finding is in agreement with the data shown for the tissue vs. all approach (Appendix A). In the remainder of the manuscript, we present data from the tissue vs. all approach for brevity and simplification.

Next, we determined which tDR subtypes were enriched in the tissue comparisons. Starting with ntDRs, the most abundant differentially expressed class in the brain vs. all comparison was i-tRFs followed by tRF3b, 5′tiRNAs, and tRF5c; meanwhile, in the spleen, the most abundant differentially expressed class was i-tRFs followed by tRF3b (Figure 4A,B). In other comparisons, either i-tRFs or tRF3s were the most significantly enriched tDR subclasses (Appendix A). The size of the upregulated tDRs in significant comparisons showed variations consistent with differences in the relative numbers of enriched tDR subclasses (Figure 4C). These differences were statistically significant when applying the Kolmogorov–Smirnov (K.S.) statistical test (Figure 4D). Mapping the significantly upregulated tDRs in the tissue vs. all comparisons to the nucleotide positions in the tRNAs showed the heterogeneity in the origin of the ntDRs and the differences in the biogenesis of tissue-specific ntDRs (Figure 4E). For example, the Sprinzl heatmap clearly showed the enrichment of upregulated 5′tiRNAs and tRF5 fragments in the brain vs. all comparison. For mtDRs, we examined those from the spleen, brain, heart, kidneys, and lung tissues vs. all comparison. i-tRFs dominated the enriched tDRs in all comparisons, except in the kidney, which showed only a handful of enriched tDRs (Figure 4F,G, Appendix A). The same patterns of enriched tDRs having size heterogeneity between tissues, evident in ntDRs analysis, were observed in the mtDRs (Appendix A).

### 2.3. tDRs Enriched in Different Tissues Have Unique tRNA Origins

To identify whether ntDRs enriched in different tissues originate from different tRNAs, we identified the isotype (amino acid decoding or isoacceptor tRNA) and the anticodon sequences of the parent tRNAs and conducted a Fisher’s exact test with Benjamin–Hochberg false discovery rate (BH-FDR) multiple test correction to identify the statistically significant odds that enriched the tDRs that originated from a specific tRNA isoacceptor (Figure 5A,B). The brain had the strongest patterns of source tRNA enrichment: we observed that upregulated ntDRs in the brain were mostly derived from Val-CAC, Val-AAC, Lys-TTT, and Ala-TGC (Figure 5A,B). Other tissues also showed varying patterns of enrichment. For example, upregulated ntDRs in the lung originated from iMet and Glu tRNAs; meanwhile, in the spleen, there was a preference towards Glu, Gly, and Leu tRNAs. We next investigated whether specific isodecoder tRNAs could contribute more to the enrichment of ntDRs (Figure 5C, Appendix A). Indeed, we observed that, for each tRNA, several isodecoders had higher odds of being the parents of the enriched ntDRs in each tissue. However, some isodecoders were more likely to be the origins of ntDRs than others; importantly, not all of the isodecoders of a given tRNA were found to be significant in our analysis. Previously, it was reported that tRNA isoacceptors are expressed uniformly in different tissues [49,53]; however, isodecoder expression was found to be varied [53]. To test whether mature tRNA isodecoder expression would vary between tissues, in the hopes of explaining the observed enrichment patterns, we re-analyzed the mature tRNA dataset from the same tissues [49] using the tissue vs. all approach to comparison (Appendix A). Only in the brain tissue did we observe some correlation between the upregulated isodecoders and the isodecoder origins of the ntDRs. However, in all other tissues, there were either no differentially expressed isodecoders, or the few that were significant did not correlate with ntDRs sources. Thus, the differential enrichment of tRNA isodecoders or isoacceptors cannot be used in explaining the differential enrichment of ntDRs in different tissues.

### 2.4. Highly Expressed Tissue-Resident tDRs Are Heterogeneous

Our analysis to this point focused on the tissue-specific tDRs, given that we analyzed the upregulated tDRs in the tissue vs. all comparisons; thus, we then asked whether the highly expressed tDRs are uniform across tissues. We extracted the top 500 ntDRs by normalized read counts and analyzed them across tissues. We observed variations in the tDRs’ subclasses (Figure 6A) and isotype sources (Figure 6B). For example, tRF5c showed higher enrichment in the spleen, while i-tRFs represented the lowest number of tDRs in the spleen but nearly 40~50% of the top tDRs in the heart and kidney. The heatmap (Figure 6C) of the top 10 expressed tDRs in each tissue and the upset plot (Figure 6D) of the top 200 expressed tDRs shows the heterogeneous expression of highly expressed tissue resident tDRs across tissues. Importantly, the spleen and lungs closely clustered together (Figure 6C), following the trend observed at multiple analysis levels that was also previously reported at the level of mRNA translation and tRNA modifications [49]. Furthermore, analysis of the isotype tRNA, isoacceptors, and isodecoders tRNA sources of the top 500 tDRs yielded varying enrichment patterns across tissues (Appendix A). The heart and brain showed enrichment in ntDRs derived from multiple tRNA-Gln-CTG isodecoders, while the spleen showed enrichment of ntDRs derived from multiple Gly-GCC isodecoders. Other patterns of isotype, isoacceptor, and isodecoder enrichment were observed across tissues. Overall, our analysis reveals significant heterogeneity in terms of the most abundant tDR expressions across different tissues as well as the source of those expressed tDRs and their biogeneses.

### 2.5. G4 Quadruplexes Forming tDRs Are Enriched in the Spleen

tDRs functions have come under significant scrutiny in recent years [54]. tDRs were proposed to interact with RBPs via motif sequences to displace them from their binding mRNAs [26], to enhance translation of ribosomal mRNAs [28,29], and to form G4 quadruplexes [17]. Given that, in RNAs, function follows form [1,35], we interrogated whether tissue resident and enriched ntDRs can form RNA G4 quadruplexes (rG4s). We used pqsfinder [55] to predict whether a given tDR is able to form G4 quadruplex or not. First, we analyzed the top 500 expressed ntDRs in each tissue. We observed varying numbers of potential G4 quadruplexes forming ntDRs in different tissues, with the highest numbers occurring in the spleen, muscles, and lungs (Figure 7A,B). However, the pqsfinder score, which predicts the strength of the potentially formed G4 structure [55], was more or less evenly distributed across tissues. We examined the isotype origin of the predicted G4 quadruplexes forming tDRs and their tDR subclass and found that most of these tDRs in all tissues originate from glutamate tRNAs (Figure 7C). The lungs, muscles, and spleen showed a population of Val-originating tDRs which were also capable of forming G4 quadruplexes; these generally did not show any contribution in other tissues, and if they did, it was only minor. tDRs originating from SeC tRNAs also contributed to the rG4-forming ntDRs in the spleen and lungs. Other tRNAs such as Asp, Leu, and Arg also contributed with varying extents in different tissues. Examining the tDR subclasses that contribute to G4 quadruplex formation revealed more heterogeneity between tissues, especially when considering spleen and lungs (Figure 7D). While most tissues showed that the G4 quadruplexes forming the ntDRs to be of the i-tRF subtype, the spleen and lungs showed more contributions to tRF3 and, to a lesser extent (but higher than other tissues), 5′tiRNAs.

Analysis of the differentially expressed ntDRs (i.e., those with specific tissue enrichment in the tissue versus all comparisons) revealed that the kidneys, heart, and spleen are specifically enriched in potential rG4-forming ntDRs (Figure 7E,F). Examining the isotype and class distribution of G4 quadruplexes forming differentially expressed ntDRs revealed a different picture compared to the most expressed ntDRs (Figure 7G,H). For example, there was a more uniform distribution of the isotype contribution to the G4-forming ntDRs in the spleen, with less dominance of the Glu-originating ntDRs (Figure 7G). tRF3 was the most dominant subclass in the rG4-forming ntDRs in the spleen; meanwhile, in kidneys and heart, i-tRFs were the dominant class (Figure 7H).

In summary, the tDRs that are highly expressed and enriched in different tissues show the potential to form G4 quadruplexes. Previously, only stress-induced 5′tiRNAs were the subject of research focusing on the G4-quadruplex-forming potential of tDRs [17,31]; however, other subclasses of ntDRs show the potential for the assembly of G4 quadruplexes. Nonetheless, this remains a computational prediction, and more biochemical and molecular analyses are needed for validation.

### 2.6. Sequencing Methods Omitting RNA Deacetylation Skew Towards 5′tiRNA Fragments

To evaluate the effect of the different steps of small RNA preparation on the final library output, we analyzed mouse tissue samples sequenced by PANDORA-seq [41] using our analysis pipeline (i.e., tDRnamer and our custom *R* script). PANDORA-seq analysis of mouse tissues included three tissues: brain (N = 3), liver (N = 3), and spleen (N = 2). For brevity, we focused on ntDRs. PANDORA-seq libraries had lower ntDRs counts compared to our libraries (Figure 1C and Figure 8A). The numbers of unique ntDRs detected per tissue were also lower in PANDORA-seq compared to our method by almost 3–10-fold (brain, 12,462 vs. 2220; liver, 10,163 vs. 1181; spleen, 2988 vs. 724) (Figure 8B,C). To determine whether these differences could be explained by sequencing depth, we compared the number of reads per input fastq file after trimming. The depth of our sequencing dataset was comparable, or even lower in the case of the liver samples, to PANDORA-seq depth (Figure 8D). Thus, the 3–10-fold increase in tDRs coverage and detection is, thus, not an issue of sequencing depth bias; rather, it reflects a drastic improvement in the library preparation method. PANDORA-seq method entails gel isolation of 15~50 nt RNAs followed by T4 PNK treatment and then demethylation before small RNA-library preparation [41]. Our method entails isolation of small RNAs (<200 nts) using commercial kit, deacetylation, demethylation, then T4 PNK end repair before small RNA-library preparation [49]. We computationally isolated the 15~50 nt tDR fragments afterwards. To evaluate how these changes impact the subclasses and distribution of detected ntDRs, we evaluated the subclass and isotype distribution of the detected ntDRs in the PANDORA-seq dataset. We observed an abundance of 5′tiRNAs in the PANDORA-seq dataset (Figure 8E), with 3′tRF fragments being underrepresented in comparison with our dataset (Figure 1E). tRNA isotype membership of ntDRs in the PANDORA-seq dataset was also less heterogeneous than our dataset, with significant underrepresentation of many tRNA isotypes (Figure 8F). Pairwise differential tDRs analysis also showed significant drifts compared to our method (Figure 8G). While we observed the spleen in our analysis to be tissue that was the most distant from all the other tissues, the PANDORA-seq analysis revealed that the brain was in fact the most distant (Figure 8G). This could be due to the abundance of 5′tiRNAs in the PANDORA-seq dataset, which we observed to be more enriched in the brain according to the brain tissue vs. all tissue comparison. The length distribution of detected ntDRs in the PANDORA-seq method showed a pronounced truncation of tDRs around 40 nts, potentially limiting the detection of longer i-tRF fragments (Figure 8H). In the brain tissue vs. all tissue analysis, we also observed the brain to have the highest number of differentially expressed ntDRs (Figure 8I). Most of the significant tDRs were of the i-tRF subclass (Figure 8J).

In summary, our analysis shows a 3–10-fold increase in tDR detection using our optimized tRNA/tDR sequencing approach in comparison with the previously used method. Such an improvement, achieved using a straightforward method that can be easily adopted, shall improve our understanding of tDRs biology and enable explorations of their origins and molecular functions beyond what was previously achieved.

## 3. Discussion

In this analysis, we demonstrate the significant heterogeneity in nuclear and mitochondrial tDRs across different tissues. Importantly, tDRs are heterogeneous at many levels, including source tRNAs, size, site of cleavage/biogenesis, subclass, structure, and potential tRNA modifications in their sequences from the source tRNAs. We also noted that, while the spleen had the lowest number of detected tDRs, it displayed the greatest diversity. tDRs have been implicated in regulating post-stroke immune responses and can replace microRNAs to regulate immune cell functions [56]. The closest tissue to spleen was found to be the lungs, which are also known for their high macrophage and immune cell populations. Source tRNAs also showed significant heterogeneity across tissues. Importantly, the expression of tRNA isoacceptors, which are stable across tissues [49,53,57], and isodecoders could not explain the patterns of tDRs enrichments. Collectively, this information indicates that the biogenesis and production of tDRs in different tissues is not a matter of simple tRNA turnover, but is a highly orchestrated process that potentially serves to regulate gene expression and the physiological functions of different tissues and cells. A potential example for this specialized functionality of various tDRs is the enrichment of potentially G4 quadruplexes forming ntDRs. G4 quadruplexes can be related to tDRs’ function via facilitating phase separation or RBPs binding [17,31,58]. The enrichment of potentially G4-quadruplex-forming ntDRs in the different tissues alludes to a potential physiological regulatory function of this subset of tDRs that was not previously studied. It is important to note that the analysis of rG4s herein remains computational and preliminary, and more research is needed to validate their presence and potential molecular roles.

An important step we achieved herein that will aid in understanding tDRs biology is the great improvement in tDR detection compared to previous methods, i.e., an increase of up to tenfold was achieved compared with previous methods [41]. This was achieved by a combination of deacetylation, demethylation, and the T4 PNK-mediated end repair of small RNAs without additional gel-purification steps. Thus, our approach is straightforward and can be readily adopted for tDR sequencing. This improved method enhances our ability to study tDR biology by enabling the discovery of previously underrepresented classes, especially tRF3s, whose amino acetylation impacted their detection in previous studies [48].

It is important to note that, to fully understand tDRs biogenesis and function, given their extreme heterogeneity, more robust datasets are needed. In addition to robust tDR profiling, combining tDR sequencing with approaches to map tRNA or tDR modifications, G4 quadruplexes formation, or potential RBPs bindings at a large scale will be necessary if we are to decipher the functions of tDRs as well as the regulation of their biogenesis. To date, the biogenesis of tDRs is not fully understood. Several enzymes apart from angiogenin contribute to tRNA cleavage and regulate mRNA translation through modulating codon-biased translation [9,11,59]. However, further work is needed to directly link specific tDRs with translational phenomena.

## 4. Methods

**Datasets:** A small RNA-seq dataset was retrieved from a previously published study [49]: sequence read archive project number, PRJNA1003133. The dataset includes small RNA-seq fastq.gz files from 7 mouse tissues with 3 biological replicates per tissue. The PANDORA-seq dataset was retrieved from Gene Expression Omnibus (GSE144666).


**Data analysis:**



**Read processing and tDR calling:**


tDRnamer, standalone version [8] (https://github.com/UCSC-LoweLab/tDRnamer, data accessed on 27 August 2025), was used to analyze the tDRs in the small RNA-seq dataset after the adaptors were trimmed and the paired-end reads were collapsed to single fastq.gz files using SeqPrep (https://github.com/jstjohn/SeqPrep, data accessed on 27 August 2025); this was conducted using max sensitivity, fastq.gz files, and mm39 database (or mitochondrial mm39 for mtDRs). tDRs with fewer than 10 reads were excluded. We used the output tDR-info.txt file for our downstream analysis. tDRnamer reports counts at a tDR sequence level, and it collapses reads that match those of multiple source tRNAs (isodecoder copies) into the same tDR ID.

To analyze tDRnamer output, we used a custom R script that first gathered the read counts for detected tDRs to create a count matrix as well as an annotation file containing all the tDRs’ information, such as their start and end positions, their sequences, and their source tRNAs. tDRs were filtered by length to ensure that tDRs with lengths between 15 and 50 nucleotides were retained for the downstream analysis.


**Normalization and differential analysis:**


EdgeR [60] was used for data analysis. Library sizes were scaled by TMM (calcNormFactors), and CPM and logCPM matrices were exported for visualization and downstream analysis. Differential enrichment was assessed per tissue versus all other tissues using a standard edgeR pipeline; significance was defined as FDR < 0.05 (Benjamini–Hochberg) and |log_2_FC| > 1.


**tDR subclass assignment:**


Each tDR was assigned to a subclass from its Sprinzl start/end: 5′ tRFs (tRF5a/b/c), 3′ tRFs (tRF3a/b, plus tRF3c when starting in the anticodon loop and not reaching the CCA), and 5′/3′ halves (tiRNA-5/tiRNA-3). tDRs covering the anticodon region and extending towards the D- and T-arms were assigned as internal tRFs (i-tRFs). Fragments were categorized as intronic when the positions were negative or when they were marked as intronic by tDRnamer. These rules are reflective of current conventions and were implemented in our R classifier.


**Sparse least square regression analysis (sPLS-DA):**


sPLS-DA analysis was conducted using the mixOmics [61] package in R using logCPM values of EdgeR as input.


**Analysis of tDRs length distribution:**


Analysis of length distribution was performed using the Sprinzl start/end information provided by tDRnamer. A weighted Kolmogorov–Smirnov test was used to compare between different conditions for statistical significance in enriched tDRs lengths.


**Sprinzl position coverage maps:**


For upregulated tDRs, each significant tDR was expanded to its covered Sprinzl positions (from the mature tRNA start to the end). We then summed the CPM over the tDRs at each position (1–76) per tissue, then row-normalized these to obtain a relative coverage (0–1) and plotted positions 1–76 on the x-axis using the ComplexHeatmap package in *R*.


**Isotype and anticodon enrichment among differentially expressed tDRs:**


For each tissue, we compared the number of significant/selected tDRs vs. the non-significant tDRs per isotype (or anticodon) against the tissue-specific background using Fisher’s exact test with BH FDR correction. We expand each tDR annotation to all its annotated source tRNAs and tested the set membership. We reported the odds ratios and FDR values and visualized the significant categories with dot-plots and heatmaps.

**Parent tRNA copy (isodecoder) enrichment**:

To resolve isodecoder specificity, each tDR was mapped to all annotated parental tRNA copies (from tDRnamer). Within each tissue, we aggregated TMM-CPM for UP vs. non-UP tDRs per copy and tested the enrichment using a CPM-weighted Fisher’s exact test and BH FDR correction. The results are shown as log_2_-odds-ratio dot-plots and −log_10_(FDR) heatmaps.


**Pairwise differential expression with edgeR quasi-likelihood (QL) omnibus F-test with post hoc contrasts:**


Raw tDR read counts were analyzed with edgeR using its quasi-likelihood (QL) generalized linear model (GLM) framework. The library sizes were normalized by the TMM method, and all models were fit to the negative binomial mean–variance family with gene-specific dispersion estimates, which were stabilized by empirical Bayes shrinkage and robust fitting.

For the multi-group comparison across tissues/conditions, we fit an intercept-based design ~ group, where the group is a factor with levels: ⟨Brain, Heart, …⟩. (Here, baseline = the first level; “Control” was set as baseline when present.) For feature i in sample j, the model was as follows:$$log\mu ij=\;logNj+\alpha i+x_{j}^{T}\beta i,$$
where μij is the expected count, *Nj* is the TMM normalization factor × library size (offset), *x_j_* contains the group indicator columns with the baseline omitted, *αi* is the intercept, and *βi* is the group effects. QL dispersions (*ϕi^QL^*) were estimated with estimateDisp/glmQLFit using robust EB shrinkage.

To determine whether any group effect was present for a given tDR, we used the omnibus QL F-test (glmQLFTest) across all non-intercept group coefficients. This provides an ANOVA-like global test per tDR that accounts for count over-dispersion and uncertainty in dispersion estimates. The resulting *p*-values were adjusted for multiple testing by the Benjamini–Hochberg procedure; we report FDR and use ⟨FDR < 0.05⟩ as significant unless noted. Heatmaps of the highest-ranked features according to the omnibus F statistic were drawn from row-z-scored log-CPM values for visualization only.

To localize the differences, we performed post hoc pairwise QL contrasts between all the unordered group pairs (e.g., Brain vs. Heart, …) with makeContrasts/glmQLFTest. For each contrast, we report log2 fold-change (logFC), *p*-value, and FDR; volcano plots are used to summarize the significance using thresholds ⟨|logFC| > 1⟩ and ⟨FDR < 0.05⟩.

Unless stated otherwise, the figures use only normalized counts for plotting; all statistical testing was performed on the original count scale within the edgeR GLM.


**G-quadruplex prediction:**


G4 quadruplexes analysis was conducted using pqsfinder [55]. We scanned tDR sequences with pqsfinder using default parameters and recorded the maximum pqsfinder score per sequence. A tDR was called “PQS-positive” if its maximum score exceeded the default significance threshold (min score = 15). We summarize both the fraction of PQS-positive tDRs and the score distribution per tissue.

All heatmaps, volcano plots, and other visualizations were created in R. Upset plots were generated using ComplexUpset package in R. Heatmaps were created using the ComplexHeatmap package in R.


**Mature tRNA analysis:**


Mature isodecoder tRNA reads were prepared using tRAX software [8]. EdgeR was used to conduct the one tissue vs. all tissue analyses for the tRNA isodecoders.

## 5. Conclusions

In conclusion, tDRs are highly heterogeneous under physiological conditions but appear to follow consistent biogenesis rules, which are tied to their specific source tRNAs. Mature tRNA expression, at either the isoacceptor level or the isodecoder level, does not appear to be a contributing factor to tDRs biogenesis. Accordingly, we ask the following questions: What are the processes regulating tDR production? How do rG4-forming tDRs regulate cellular functions? Such questions, and many more, need to be addressed in future research. Such research will help us to fully comprehend tDR biology, support us in understanding their roles in the many diseases that they have been reported to play a role in, and potentially aid us in developing novel tDR-based therapeutics. We believe our tDR-sequencing method will support the better discovery of tDRs and assist us in developing an understanding of their biology.

## Figures and Tables

**Figure 1 ijms-26-08772-f001:**
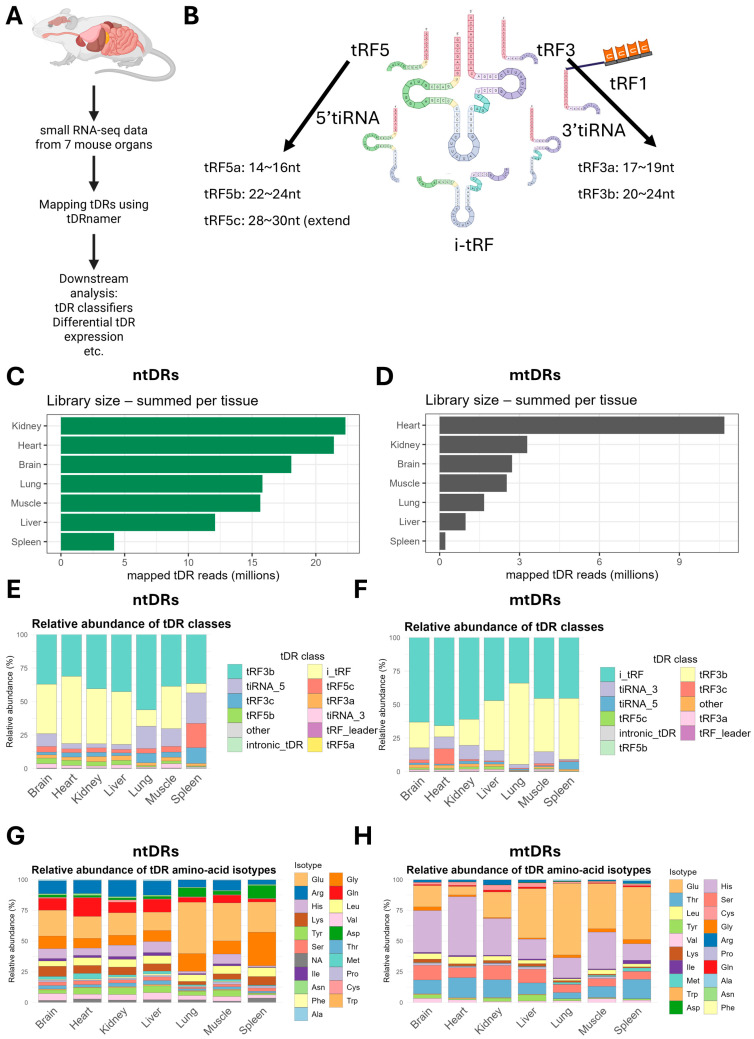
Global patterns of tDRs expression in mouse tissues. (**A**) Strategy for analyzing tDRs in mouse tissues using tDRnamer. tRNA/tDR-optimized small RNA sequencing dataset encompassing 7 mouse tissues (brain, heart, lungs, kidneys, liver, spleen, and muscle, N = 3 per tissue) [49] was used for the analysis. (**B**) The known subtypes of tDRs used for the classifier in this study. (**C**) Library size (in million reads) of the detected nuclear tDRs (ntDRs) per tissue (total number of mapped tDR reads that passed the filters and summed across samples). (**D**) Library size of the detected mitochondrial tDRs (mtDRs) in different tissues. (**E**,**F**) Relative distribution of different ntDRs (**E**) and mtDRs (**F**) subtypes/classes in different tissues. (**G**,**H**) Relative distribution of isotype of origin (i.e., amino acids decoding tRNA) of ntDRs (**G**) and mtDRs (**H**) across tissues.

**Figure 2 ijms-26-08772-f002:**
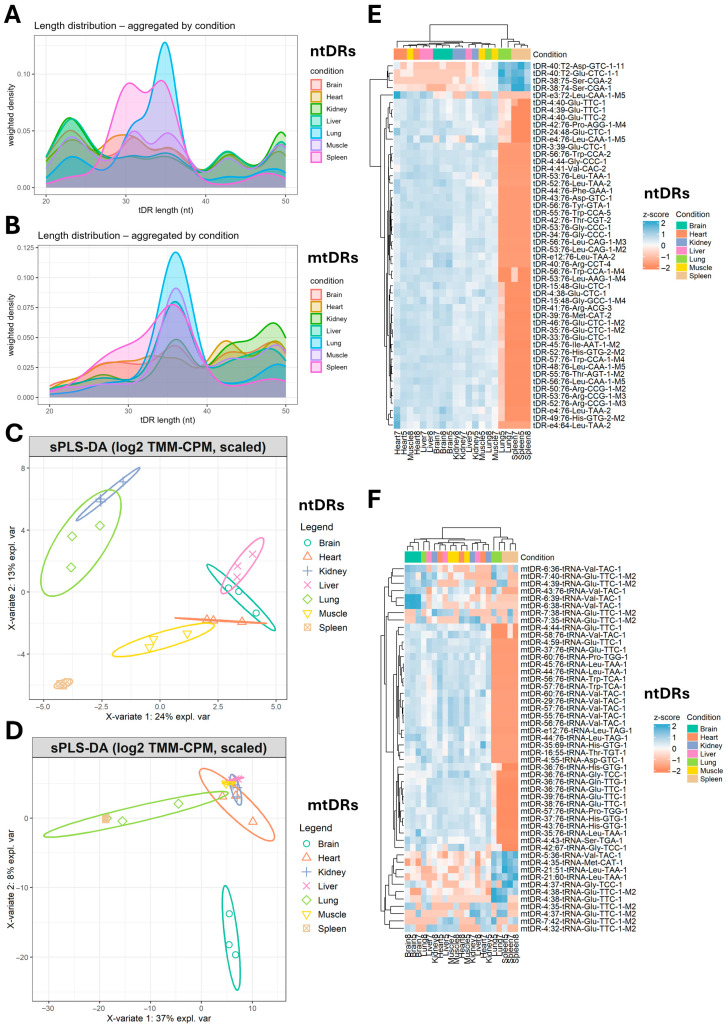
(**A**) Length distribution of ntDRs across tissues. (**B**) Length distribution of mtDRs across tissues. (**C**) Sparse partial least square discriminant analysis (sPLS-DA) clustering of ntDRs. (**D**) sPLS-DA clustering of mtDRs. (**E**) Heatmap of top variable ntDRs. (**F**) Heatmap of top variable mtDRs.

**Figure 3 ijms-26-08772-f003:**
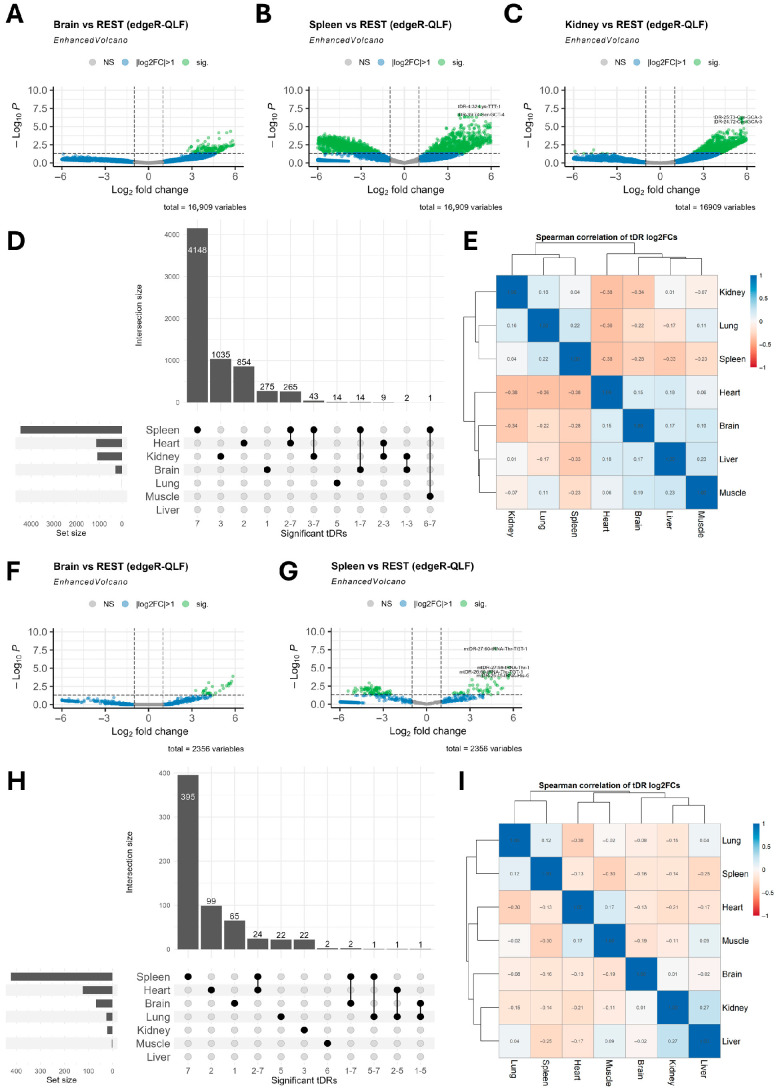
(**A**–**C**) Volcano plot of differentially expressed ntDRs in different tissues. Each volcano represents a tissue vs. all other tissues comparison. Significance cutoff of differentially expressed tDRs was |log2FC| > 1 and FDR < 0.05. (**D**) Upset plot of differentially expressed ntDRs showing the uniquely expressed ntDRs in different tissues. The spleen had the highest number of uniquely expressed ntDRs. (**E**) Spearman’s correlation analysis of differentially expressed ntDRs using their log2 fold change (log2FC) values. The legend bar indicates Spearman’s Rho values. (**F**,**G**) Volcano plots of differentially expressed mtDRs in different tissues. (**H**) Upset plot of differentially expressed mtDRs showing the spleen with the highest number of uniquely expressed mtDRs. (**I**) Spearman’s correlation analysis of differentially expressed mtDRs using their log2FC values.

**Figure 4 ijms-26-08772-f004:**
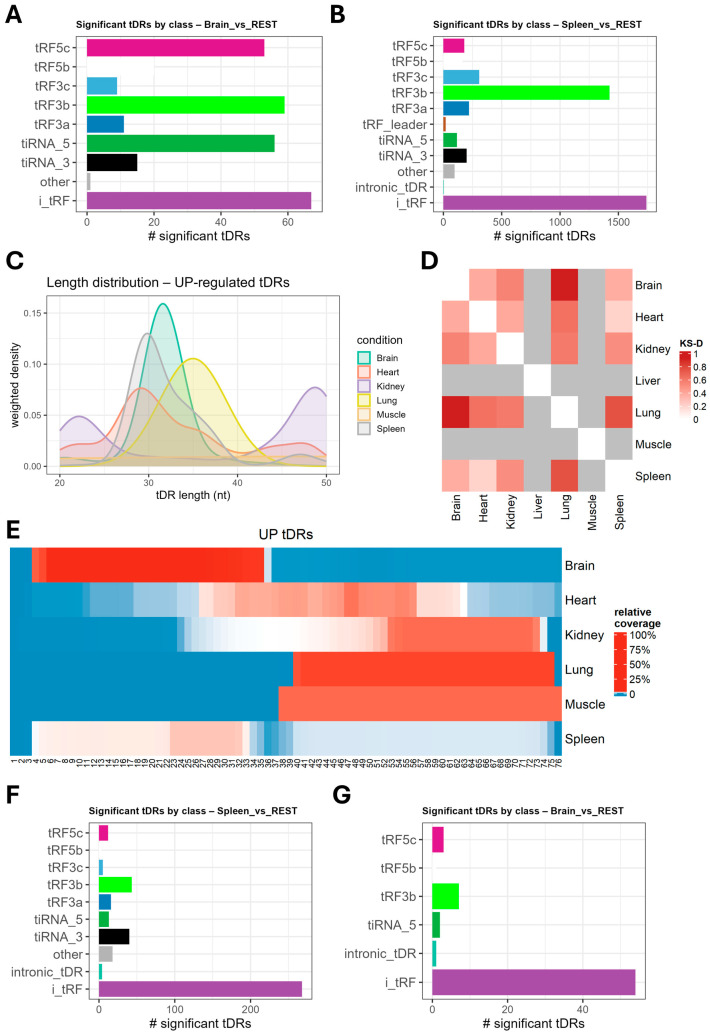
Analysis of the length distribution and subclasses of tissue-specific tDRs. (**A**) Distribution of tDR subclasses in the upregulated ntDRs in the brain vs. rest analysis. (**B**) Distribution of tDR subclasses in the upregulated ntDRs in the spleen vs. rest analysis. (**C**) Size distribution of the upregulated ntDRs in each condition vs. rest comparison. (**D**) Heatmap of Kolmogorov–Smirnov statistical analysis of length distribution of upregulated ntDRs in each condition. Higher values indicate statistical significance. All comparisons were statistically significant (*p* < 0.05, FDR < 0.05). (**E**) Sprinzl heatmap showing the mapping of significantly upregulated ntDRs across the mature tRNA sequences. The heatmap reveals the heterogeneity in tDR subclasses as well as the size distribution differences between tissues. (**F**) Subclasses of enriched mt-tDRs in the spleen vs. other tissues comparison. (**G**) Subclasses of enriched mt-tDRs in the brain vs. other tissues comparison.

**Figure 5 ijms-26-08772-f005:**
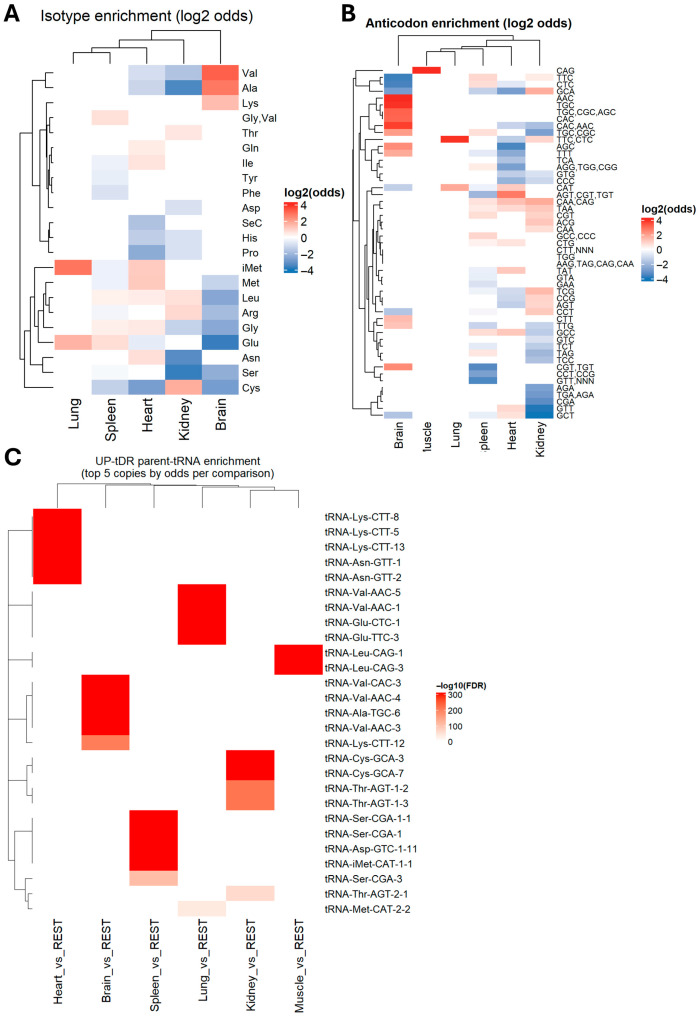
Tissue-enriched tDRs have different mature tRNA origins. (**A**) Heatmap of log2 odds ratio of isotype enrichment of upregulated ntDRs. This heatmap indicates whether an isotype tRNA has higher (or lower) odds of producing tDRs in each tissue. (**B**) Heatmap of log2 odds ratio of anticodon enrichment of upregulated ntDRs. This heatmap gives information regarding a given tRNA anticodon (i.e., isoacceptors) being the source of ntDRs in each tissue. (**C**) Heatmap of −log10 FDR of parent tRNA enrichment of ntDRs. This heatmap gives information regarding the mature tRNA source of upregulated ntDRs. Importantly, this analysis gives information about whether specific isodecoders are the sources of tDRs. The heatmap represents the top 5 tRNAs in each tissue by odds ratio. Expanded plots showing the log2 odds ratio results of the top 25 isodecoder tRNA tDR sources can be found in the Appendix A.

**Figure 6 ijms-26-08772-f006:**
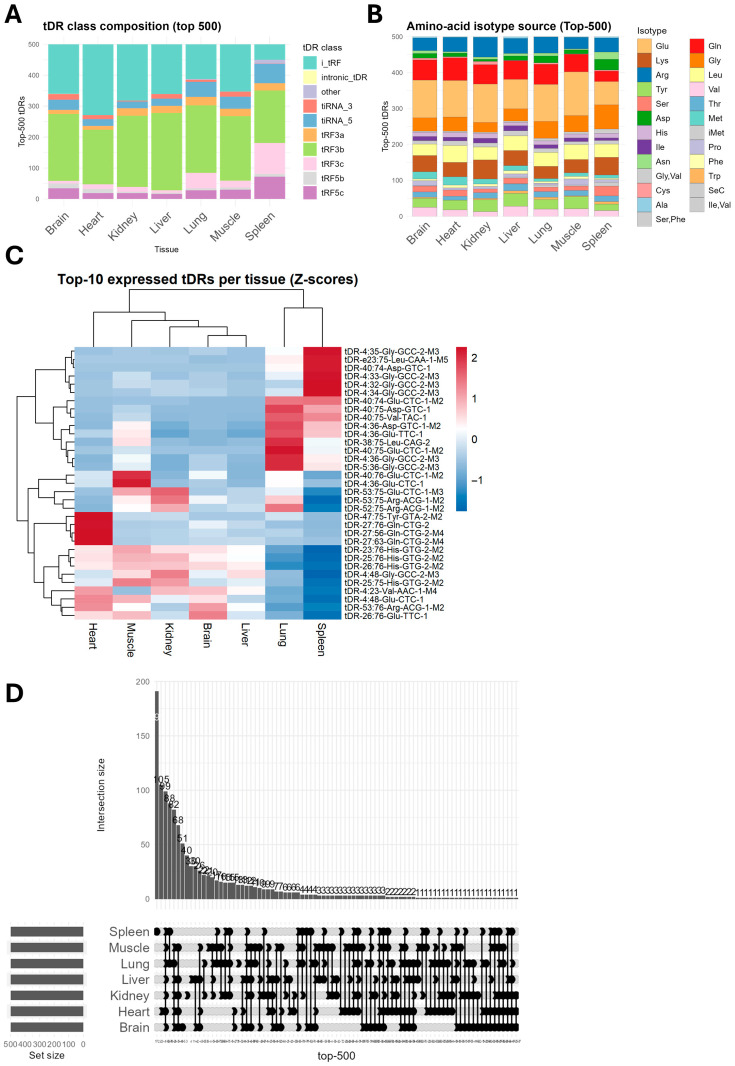
The most expressed tDRs are heterogeneous across tissues. (**A**) Subclasses of highly expressed (top 200 by normalized read counts) ntDRs in different tissues. (**B**) Isotype classes of highly expressed ntDRs in different tissues. (**C**) Heatmap of the top 10 expressed ntDRs in each tissue. (**D**) Upset plot of the top 200 expressed ntDRs showing overlapping and unique ntDRs across tissues.

**Figure 7 ijms-26-08772-f007:**
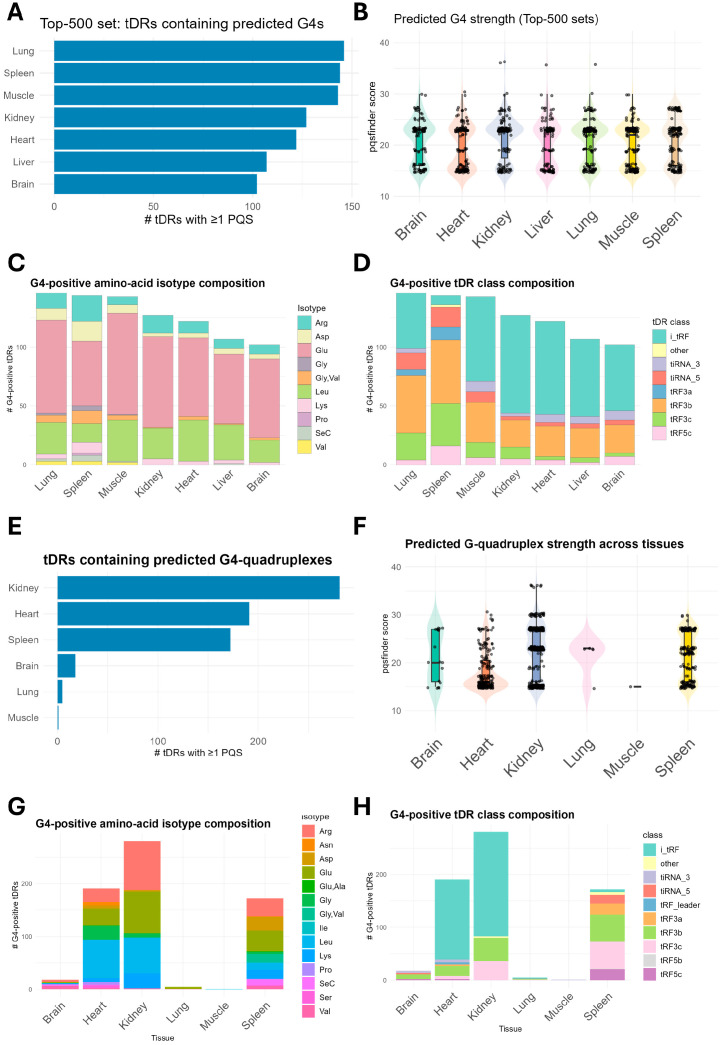
ntDRs can form G4 quadruplexes with special enrichment of G4-forming tDRs in the spleen. (**A**) Bar plot showing the numbers of ntDRs predicted to be able to assemble into G4 quadruplexes in each tissue from the top 200 expressed ntDRs (top ntDRs). (**B**) pqsfinder score of the top potentially G4-forming ntDRs in each tissue. (**C**) Isotypes of the top G4-forming ntDRs in each tissue. (**D**) tDR subclasses of the top G4-forming ntDRs. (**E**) Bar plot representing the numbers of predicted G4-forming ntDRs in the tissue enrichment analysis (i.e., differentially expressed ntDRs). (**F**) pqsfinder score of the potentially differentially expressed G4-forming ntDRs in each tissue. (**G**) Isotypes of the differentially expressed G4-forming ntDRs in each tissue. (**H**) tDR subclasses of the differentially expressed G4-forming ntDRs.

**Figure 8 ijms-26-08772-f008:**
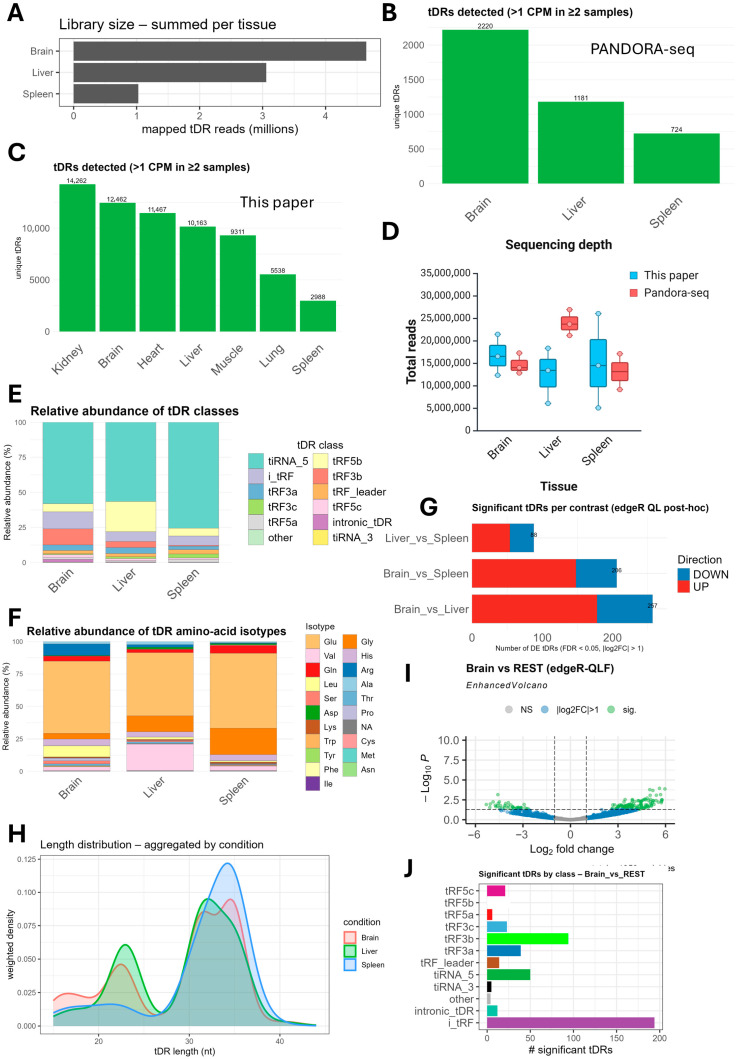
Benchmarking our method against PANDORA-seq method reveals a 3–10-fold increase in ntDRs detection efficiency. (**A**) Library size (in million reads) of the detected nuclear tDRs (ntDRs) per tissue (total number of mapped tDR reads that passed the filters and summed across samples). (**B**) Numbers of uniquely detected ntDRs in different tissues in PANDORA-seq dataset. (**C**) Numbers of uniquely detected ntDRs in our dataset. (**D**) Comparison of sequencing depth between PANDORA-seq dataset and our dataset. FastQC was used to identify the number of reads per trimmed fastq.gz file. (**E**) Relative distribution of different ntDRs subtypes/classes in different tissues in the PANDORA-seq dataset. (**F**) Relative distribution of isotype of origin (i.e., amino acids decoding tRNA) of ntDRs across tissues in the PANDORA-seq dataset. (**G**) Bar plot of the number of significant tDRs on pairwise differential expression analysis between the 3 tissues in the PANDORA-seq dataset. (**H**) Length distribution of the detected ntDRs in the PANDORA-seq dataset. (**I**) Brain tissue vs. other tissue: differential ntDRs expression analysis of the PANDORA-seq dataset. (**J**) The contribution of different subclasses to the differentially expressed ntDRs in the brain tissue vs. other tissue analysis of the PANDORA-seq dataset.

## Data Availability

The small RNA-seq dataset was retrieved from a previously published study [49], sequence read archive project number: PRJNA1003133. The PANDORA-seq dataset [41] was retrieved from Gene Expression Omnibus (GSE144666).

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
