# Peer review of "Optimized tDR Sequencing Reveals Diversity and Heterogeneity in tRNA-Derived Fragment Landscapes in Mouse Tissues"

_ijms, 2025, doi:10.3390/ijms26188772_

Round 1

Reviewer 1 Report

Comments and Suggestions for Authors

This manuscript presents a comprehensive atlas of tRNA-derived small RNAs (tDRs) across seven adult mouse tissues, using a small RNA sequencing dataset optimized for tDR detection. The study identifies over 26,000 nuclear tDRs and 5,000 mitochondrial tDRs, revealing strong tissue-specific heterogeneity in abundance, subclass composition, origin, and structural potential.

The prediction that certain tDRs form G-quadruplex structures is intriguing and raises the possibility of novel immune-related functions. However, the sequencing library preparation method for tDRs has already been established, and the current bioinformatics analyses and predictions alone may not be sufficient to fully support the claim of “selective biogenesis” or to elucidate the biological significance of G-quadruplex structures. I recommend that the authors conduct additional molecular experiments and further analyses to clarify and strengthen their arguments.

Major Concerns

  1. Mechanisms of Selective Biogenesis
    The study convincingly demonstrates that tDR enrichment patterns cannot be explained by mature tRNA abundance, supporting the idea of selective cleavage. However, the analysis and discussion could be expanded to consider enzymatic or modification-driven mechanisms. For example, Angiogenin, Dicer, RNase Z, SLFN family members, and tRNA modification enzymes (e.g., NSun2, ALKBH1) are known to regulate tRNA cleavage. The expression patterns of these factors have been studied in multiple tissues. I encourage the authors to analyze and discuss correlations between these genes and the observed tDR patterns, as this could provide mechanistic insights into the selective expression of tDRs.

  1. G-Quadruplex Structures
    The enrichment of predicted G4-forming tDRs in the spleen is an interesting and potentially significant finding. However, the conclusions currently rely entirely on in silico prediction without experimental validation, which weakens the impact. The authors should provide experimental evidence that at least some of these tDRs indeed form G-quadruplex structures and explore their potential biological roles, particularly in immune-related functions (e.g., through biophysical assays or functional studies using G4 ligands).

Minor Concerns

  1. Please ensure consistent terminology (e.g., “heterogeneous” rather than “heterogenous”).
  2. Some abbreviations (e.g., tiRNA for tRNA-derived stress-induced RNA) should be spelled out at first mention for clarity.

Author Response

We thank the reviewer for his comments. Below is a point-by-point response:

The prediction that certain tDRs form G-quadruplex structures is intriguing and raises the possibility of novel immune-related functions. However, the sequencing library preparation method for tDRs has already been established, and the current bioinformatics analyses and predictions alone may not be sufficient to fully support the claim of “selective biogenesis” or to elucidate the biological significance of G-quadruplex structures. I recommend that the authors conduct additional molecular experiments and further analyses to clarify and strengthen their arguments.

Response: While some methods have been established for tDR sequencing, we show in our updated revised manuscript that we were able to greatly improve the detection of tDRs in sequencing libraries, up to 10 folds compared to previously published datasets using the same mouse tissue analysis approach and at comparable sequencing depth (see new figure 8). Our method also robustly detect the under represented tRF3 fragments compared to other methods that were skewed towards 5’ and internal fragments. Our work herein aims to show that tDRs are heterogeneously expressed in healthy tissues, and that they are not random tRNA degradation products. Future works will build on this analysis to explore further the understudied role of tDRs in normal physiology.

Major Concerns

  1. Mechanisms of Selective Biogenesis
    The study convincingly demonstrates that tDR enrichment patterns cannot be explained by mature tRNA abundance, supporting the idea of selective cleavage. However, the analysis and discussion could be expanded to consider enzymatic or modification-driven mechanisms. For example, Angiogenin, Dicer, RNase Z, SLFN family members, and tRNA modification enzymes (e.g., NSun2, ALKBH1) are known to regulate tRNA cleavage. The expression patterns of these factors have been studied in multiple tissues. I encourage the authors to analyze and discuss correlations between these genes and the observed tDR patterns, as this could provide mechanistic insights into the selective expression of tDRs.

 Response: We agree with the reviewer that many enzymes have been reported to be involved in tDRs biogenesis. However, in most cases these enzymes were studied under specific conditions, mostly pathological. For example, SLFN family are reported in conditions of DNA damage or infection and so on. We opted not to discuss these enzymes in this work, as we would rather focus on tDRs themselves in our discussion for brevity and focus.

  1. G-Quadruplex Structures
    The enrichment of predicted G4-forming tDRs in the spleen is an interesting and potentially significant finding. However, the conclusions currently rely entirely on in silico prediction without experimental validation, which weakens the impact. The authors should provide experimental evidence that at least some of these tDRs indeed form G-quadruplex structures and explore their potential biological roles, particularly in immune-related functions (e.g., through biophysical assays or functional studies using G4 ligands).

Response: We have updated our analysis and show the detection of rG4 forming tDRs in multiple tissues. Currently, most work done on rG4 tDRs is done in vitro using cell models to the best of our knowledge. While certain methods could be adapted and optimized for rG4 analysis from tissues, we believe those would need dedicated research efforts and not as a small part of a project. In fact, our aim was to stimulate the interest of scientists in potential tDR functions through this in silico analysis of rG4 forming capacity. We clearly state that this is an in silico analysis and not a definitive proof and indeed more work is needed to explore such patterns.

Minor Concerns

  1. Please ensure consistent terminology (e.g., “heterogeneous” rather than “heterogenous”).

Response: We corrected this throughout the manuscript

  1. Some abbreviations (e.g., tiRNA for tRNA-derived stress-induced RNA) should be spelled out at first mention for clarity.

Response: We corrected this throughout the manuscript

Reviewer 2 Report

Comments and Suggestions for Authors

The authors performed a follow-up analysis of their previously published dataset on a different angle, which is the expression of tRNA-derived RNA study in seven mouse tissues. The use of ARM-seq and PNK treatment made these data more reliable in assessing tDR expression than traditional small RNA-seq protocols. The identification of the putative G-quadruplex in tissue-specific tDRs is also a novel finding. However, the article has the following limitations that should be addressed:

  1. Although many of the previous studies were performed using traditional small RNA-seq protocols, some of them such as the mouse atlas in Isakova et al. (2020) are highly relevant and should be compared with the current study.
  2. The authors used DESeq2 to normalize tDR read counts. However, the high variation of the tDR read counts among the tissue types (especially the relatively low abundance in spleen) violates the core assumptions of DESeq2 which was designed for balanced experimental designs with similar library sizes, making misleading or biased normalized results. The authors should try out other normalization strategies such as EdgeR’s TMM or zero-inflated models to handle these data.
  3. This study applied a “tissue vs all other” approach to assess the tDR enrichment in each tissue type. This means biologically disparate tissues were pooled together as a single heterogeneous control group, which increases variance and reduces statistical power to detect true tissue-specific differences, especially when there is high variation of the tDR abundance among the tissue types. This leads to uncertainty of whether the differences in the comparisons reflect the target tissue's uniqueness or the dissimilarity to the pooled control. The authors should instead consider other approaches such as all pairwise comparisons with FDR corrections or ANOVA-based approaches.

Author Response

We thank the reviewer for their important and insightful comments that helped us to greatly improve our manuscript and workflow. We provide here a point-by-point response to their concerns:

  1. Although many of the previous studies were performed using traditional small RNA-seq protocols, some of them such as the mouse atlas in Isakova et al. (2020) are highly relevant and should be compared with the current study.

Response: The study by Isakova et al is certainly one of the important studies that we found that attempted to interrogate small RNAs expression across mouse tissues. However, the differences in sequencing approaches made direct comparisons inappropriate. Mainly, the library preparation in Isakova et al study was done using illumina TruSeq small RNA library preparation kit. Isakova et al reported no special treatment for their small RNAs, such as the commonly used demethylation and end-repair, which are necessary for proper detection of tDRs. In the dataset we used, the libraries were prepared using 3 pre-treatment steps: Deacetylation, demethylation, then end repair, before the libraries were prepared. The deacetylation step is essential for detection of 3’tRFs. It is clear in Isakova et al’s work (see figure 4C in their article) that the representation of 3’tRFs in nuclear tDRs is very low compared to our study (see figure 1E). Thus, direct comparison with such huge differences in output would be inappropriate. It is now understood that a minimum of demethylation and end-repair is needed for good coverage of tDRs (see PANDORA-seq method in Shi et al, Nat Cell Bio, 2021.  Figure 2 shows the differences in coverage of tDRs with each treatment step). However, we expanded our analysis to include the mouse tissue analysis conducted in the PANDORA-seq paper (Shi et al). Given that the difference between our method and PANDORA-seq is in how small RNAs are purified and in the deacetylation step, it was appropriate to compare these 2 methods. We show that our method can detect 3~10 fold more tDRs at the same or lower sequencing depth compared to PANDORA-seq. We expect the difference to be even greater when compared to Isakova et al which did not utilize any pretreatment for tDRs detection.

  1. The authors used DESeq2 to normalize tDR read counts. However, the high variation of the tDR read counts among the tissue types (especially the relatively low abundance in spleen) violates the core assumptions of DESeq2 which was designed for balanced experimental designs with similar library sizes, making misleading or biased normalized results. The authors should try out other normalization strategies such as EdgeR’s TMM or zero-inflated models to handle these data.

Response: We thank the reviewer for pointing out our oversight. Indeed, EdgeR TMM is better suited to our dataset. We have reworked our pipeline to use EdgeR’s TMM normalization and conducted all differential tDR expression analysis using EdgeR. We observed more differentially expressed tDRs after using EdgeR, which improved our downstream analysis. There have been some minor differences after changing the workflow, and we updated our manuscript accordingly. However, the overall conclusions and assumptions did not change.

  1. This study applied a “tissue vs all other” approach to assess the tDR enrichment in each tissue type. This means biologically disparate tissues were pooled together as a single heterogeneous control group, which increases variance and reduces statistical power to detect true tissue-specific differences, especially when there is high variation of the tDR abundance among the tissue types. This leads to uncertainty of whether the differences in the comparisons reflect the target tissue's uniqueness or the dissimilarity to the pooled control. The authors should instead consider other approaches such as all pairwise comparisons with FDR corrections or ANOVA-based approaches.

Response: We have conducted multi-pairwise comparisons with quasi-likelihood analysis using EdgeR to identify differentially expressed tDRs and tissue specific enrichment patterns. In that analysis, only in the spleen could we identify 188 specific ntDRs, but we could only identify a handful of tissue specific tDRs in other tissues or none. Ideally, we would use multinomial modeling to identify those tissue specific tDRs. While we have created a script to do that, the small number of biological replicates (3 in our case) rendered the model underpowered. We show the pairwise EdgeR analysis with omnibus QL F-test in supplementary figure 2. 

Reviewer 3 Report

Comments and Suggestions for Authors

Summary:

The manuscript entitled “Selective Biogenesis and Structural Diversity Shape tRNA-Derived Fragment Landscapes across Mouse Organs” by the authors Ando et al., investigate the occurrence and heterogeneity of tRNA-derived fragments (tDRs). The authors use their previously published dataset generated from 7 different mouse tissues to investigate tDR tissue specificity. The authors reveal tissue specific tDR enrichments. Additionally, the authors investigate tDR origin and reveal potential G-quadruplex structural features. However, some methodological details, such as alignment strategies, are insufficiently described or discussed. Moreover, the lack of dataset comparisons, and normalization for parent tRNA abundance, making it difficult to fully evaluate the robustness and interpretation of the findings.

Minor:

  1. The title does not accurately reflect the scope of the study. While it implies a broader investigation of structural diversity, the analysis is restricted to potential G4 structures. Similarly, the use of the term “selective biogenesis […] shape tRNA-Derived Fragment Landscape [..]” is misleading. The authors measure global tDR levels and analyze heterogeneity, but they do not directly investigate the mechanisms of biogenesis or how these processes shape tDR diversity. Hence, the title needs rephrasing.
  2. Line: Line 98-99: The authors state that RNA modification can bias tDR coverage, hence why their library preparation strategy includes demethylation. However, the paper they’re citing also includes T4 PNK treatment, to overcome ligation limitations introduced by 3’ terminal modifications (e.g. 3’-p or 2’,3’-cP). Although this manuscript purely focuses on the computational analysis and uses the data generated in their previous manuscript [49], I’m wondering why only demethylation was applied and whether these modifications still introduce sequencing biases.
  3. Line 102: The authors state that tDRnamer also uses “other” information that enables in-depth analysis. However, it is unclear what this “other” information refers to. The manuscript should clearly describe how tDRs are detected and how the analyses were performed. Moreover, the manuscript lacks critical details which are necessary to accurately evaluate the robustness of the findings. Information on how many replicates were included in the analysis, whether sequencing depth was comparable across tissues, and whether normalization was applied, are lacking and should be addressed.
  4. Figure 1: starting from Figure 1E, the font sizes are too small. Please check and adjust font sizes in all figures of the manuscript.
  5. Figure 1E, F: Please add in figure which plot is ntDRs and which plot represents mtDRs.
  6. Figure G, H: Y-axis title unclear, please revise. Additionally, it is not immediately clear whether this plot is showing tRNA isotype source of all tDRs or a specific subtype?
  7. Figure 2A - D: Please specify in the figure, which figures correspond to ntDRs and mtDRs.
  8. Line 236: I’m struggling to understand this sentence, “by copy number the uniform across tissues” seems to be a grammar issue. Please revise.
  9. Figure 6B: There seems to be an issue with the figure as I cannot distinguish the different tRNA isotope colors, since the colors are “bleeding” into each other (?) Please correct.

Major:

  1. In addition to minor comment #2: the authors further use this as an argument why other previously generated small RNA sequencing datasets were not analyzed. However, the publication they’re citing [41] which is also the basis of the method of the authors previous publication, generated datasets for mouse brain, spleen and liver. For a purely computational study examining global tDR landscapes, it would be both valuable and appropriate to incorporate comparisons between both datasets. In particular, it would strengthen the manuscript to assess the extent of overlap across datasets and to determine whether similar tDR subtypes are enriched in specific mouse tissues across datasets.
  2. Figure 1C, D: The manuscript does not clearly explain how mapping was performed. Given that tDRs often consist of halves or even smaller fragments, accurate alignment can be challenging. How is alignment quality assessed, and how might potential misalignment impact the analysis? Are the authors restricting their analysis to high-confidence reads? Additionally, how are multimapping reads handled?
  3. Line 123 – 126: The authors report that certain tRNA isotypes represent a considerable source of tDRs. However, it is unclear whether this finding reflects a genuine biological difference or simply might be caused by differences in overall tRNA abundance between tissues. Could the data be normalized to account for the relative abundance of individual tRNA isotypes?
  4. Supplementary Figures: The Figure legends overall lack critical descriptions. Please revise all descriptions.
  5. Figure 5C: I am not convinced that the current figure effectively shows the authors’ point about differences of tDR origin due to different isodecoder “parents”. As of right now, the plot does not show clear visual differences between isodecoders, which could be misleading. Supplementary Figure 3 provides a clearer representation and might be more appropriate. In addition, it is unclear how robust these observed differences are, as no statistical tests or significance values are provided in either the main figure or the supplementary material. Please revise accordingly.
  6. Results part 2.4: Building on my earlier comment regarding the relationship between tRNA expression and tDR levels, I wonder whether the observed heterogeneity in tDR expression might partly reflect differences in the expression levels of the parent tRNAs. Hence, I’m wondering if the statement that there is differences in tDR biogenesis holds true (line 251).
  7. Section 2.5: It is unclear to me how confident the G-quadruplex formation analysis is. I’m also wondering how enriched are individual tDRs for G-stretches, and how does this enrichment correlate with the pqsfinder scores predicting rG4 formation? Additionally, I’m curious about whether similar rG4 formation is predicted in the parent tRNAs. Is this something only happening in the tDRs or is this an inherited structural feature from parent tRNAs?

Author Response

We thank the reviewer for their thorough and detailed comments that helped us greatly to improve our work. We provide a point-by-point response to their concerns here:

Minor:

  1. The title does not accurately reflect the scope of the study. While it implies a broader investigation of structural diversity, the analysis is restricted to potential G4 structures. Similarly, the use of the term “selective biogenesis […] shape tRNA-Derived Fragment Landscape [..]” is misleading. The authors measure global tDR levels and analyze heterogeneity, but they do not directly investigate the mechanisms of biogenesis or how these processes shape tDR diversity. Hence, the title needs rephrasing.

Response: We have changed the title to: Optimized tDR sequencing reveals diversity and heterogeneity in the tRNA-Derived Fragment Landscapes across Mouse Organs

  1. Line: Line 98-99: The authors state that RNA modification can bias tDR coverage, hence why their library preparation strategy includes demethylation. However, the paper they’re citing also includes T4 PNK treatment, to overcome ligation limitations introduced by 3’ terminal modifications (e.g. 3’-p or 2’,3’-cP). Although this manuscript purely focuses on the computational analysis and uses the data generated in their previous manuscript [49], I’m wondering why only demethylation was applied and whether these modifications still introduce sequencing biases.

Response: We are unclear what the reviewer’s point is exactly here. In the dataset we used, which is also published from our group, 3 steps of preparation were employed: Deacetylation, demethylation, then end-repair. All these steps serve to enhance the detection of tDRs. We have detailed this information in our manuscript. These are the current available methods for preparing tRNA/tDR samples for sequencing. We are unsure what other modifications the reviewer is referring to (maybe anticodon modifications?). However, there are no other means to strip tRNAs from other modifications. We have also conducted comparative analysis between our dataset and a previously published dataset (Shi et al, Nat Cell Bio, 2021) to show the impact of tDRs pretreatment on detection.

  1. Line 102: The authors state that tDRnamer also uses “other” information that enables in-depth analysis. However, it is unclear what this “other” information refers to. The manuscript should clearly describe how tDRs are detected and how the analyses were performed. Moreover, the manuscript lacks critical details which are necessary to accurately evaluate the robustness of the findings. Information on how many replicates were included in the analysis, whether sequencing depth was comparable across tissues, and whether normalization was applied, are lacking and should be addressed.

Response: We have updated this sentence to explain what other information there is. We have also updated the methods section. As for sequencing depth, Figure 1 clearly shows the library size for ntDRs and mtDRs and tissue variations. Normalization was done using EdgeR TMM-CPM. Previously we used Deseq2 for normalization and differential expression, however, reviewer 1 pointed out the limitation of using Deseq2. The number of replicates was clearly mentioned in figure 1 legends. We added the information again in the first paragraph of the results and in the methods section.

  1. Figure 1: starting from Figure 1E, the font sizes are too small. Please check and adjust font sizes in all figures of the manuscript.

Response: We have improved the font size for all figures in the manuscript.

  1. Figure 1E, F: Please add in figure which plot is ntDRs and which plot represents mtDRs.

Response: We have highlighted which is which in the figures

  1. Figure G, H: Y-axis title unclear, please revise. Additionally, it is not immediately clear whether this plot is showing tRNA isotype source of all tDRs or a specific subtype?

Response: We have updated the figures as stated above. This figure shows the percentage of tDRs that were mapped to different tRNA isotypes. It is stated in the results that this distribution is pertaining to all tDRs detected in the dataset.

  1. Figure 2A - D: Please specify in the figure, which figures correspond to ntDRs and mtDRs.

Response: We have highlighted which is which in the figures

  1. Line 236: I’m struggling to understand this sentence, “by copy number the uniform across tissues” seems to be a grammar issue. Please revise.

Response: This was a typing error. It was corrected.

  1. Figure 6B: There seems to be an issue with the figure as I cannot distinguish the different tRNA isotope colors, since the colors are “bleeding” into each other (?) Please correct.

Response: We updated this figure

Major:

  1. In addition to minor comment #2: the authors further use this as an argument why other previously generated small RNA sequencing datasets were not analyzed. However, the publication they’re citing [41] which is also the basis of the method of the authors previous publication, generated datasets for mouse brain, spleen and liver. For a purely computational study examining global tDR landscapes, it would be both valuable and appropriate to incorporate comparisons between both datasets. In particular, it would strengthen the manuscript to assess the extent of overlap across datasets and to determine whether similar tDR subtypes are enriched in specific mouse tissues across datasets.

Response: We analyzed the PANDORA-seq dataset that the reviewer mentioned. In this dataset, there were 3 tissues analyzed: Brain (N = 3), Liver (N = 3), and spleen (N = 2). Our dataset had equal or lower absolute sequencing depth (i.e. number of reads per trimmed fastq file), however, we observed 3~10 fold more ntDRs detected in our analysis, indicating the great improvement of our method compared to previously published methods. Importantly, as predicted, there was under representation of tRF3 fragments in the previous methods as they did not perform deacetylation of small RNAs. We thank the reviewer for guiding us towards conducting this analysis as it improved the impact of our work drastically.  The new data are in figure 8 and section 2.6. We have updated our discussion and abstract accordingly.

  1. Figure 1C, D: The manuscript does not clearly explain how mapping was performed. Given that tDRs often consist of halves or even smaller fragments, accurate alignment can be challenging. How is alignment quality assessed, and how might potential misalignment impact the analysis? Are the authors restricting their analysis to high-confidence reads? Additionally, how are multimapping reads handled?

Response: tDRnamer is a published and known software from the same research group that created the Genomic tRNA Database. Information on tDRnamer mapping can be found in its documentation https://trna.ucsc.edu/tDRnamer/docs/standalone/. We have updated the methods section with additional explanations.

  1. Line 123 – 126: The authors report that certain tRNA isotypes represent a considerable source of tDRs. However, it is unclear whether this finding reflects a genuine biological difference or simply might be caused by differences in overall tRNA abundance between tissues. Could the data be normalized to account for the relative abundance of individual tRNA isotypes?

Response: We (Ando et al, IJMS, 2025) and others (Gao et al, Nat Cell Biol, 2024) have demonstrated that tRNA isoacceptors (i.e. isotypes) are stably expressed. On the other hand, Gao et al and Pinkard et al (Nat Comm, 2020) revealed that the isodecoders do vary in their ratios between tissues and cell states. Collectively, these data indicate that the observations are not due to variations in tRNA isoacceptors expression between tissues but rather in tDRs biogenesis dynamics.  We have also included Tissue vs All comparison of mature tRNAs isodecoders in Supplementary figure 6. There were minimal or no changes in isodecoder expression across tissues. Only in the brain did we observe some correlation between the few upregulated isodecoders and the upregulated ntDRs memberships. Given that other tissues did not show such correlations or patterns, it is unlikely that tRNA isodecoder expression could globally explain tDRs expression.  

  1. Supplementary Figures: The Figure legends overall lack critical descriptions. Please revise all descriptions.

Response: We expanded the figure legends as instructed

  1. Figure 5C: I am not convinced that the current figure effectively shows the authors’ point about differences of tDR origin due to different isodecoder “parents”. As of right now, the plot does not show clear visual differences between isodecoders, which could be misleading. Supplementary Figure 3 provides a clearer representation and might be more appropriate. In addition, it is unclear how robust these observed differences are, as no statistical tests or significance values are provided in either the main figure or the supplementary material. Please revise accordingly.

Response: Figure 5C is a summary of the observations highlighted in supplementary figure 3. Figure 5C uses FDR values while Supplementary figure 3 shows the odds ratio of the significant hits. We used Fisher exact test with BH FDR correction, clearly stated in the figures. The details of the statistical approach are presented in the Methods section.

  1. Results part 2.4: Building on my earlier comment regarding the relationship between tRNA expression and tDR levels, I wonder whether the observed heterogeneity in tDR expression might partly reflect differences in the expression levels of the parent tRNAs. Hence, I’m wondering if the statement that there is differences in tDR biogenesis holds true (line 251).

Response: As shown in Supplementary figure 5, and as discussed in our manuscript based on our previous work (Ando et al, IJMS, 2025) and others as well (Gao et al, Nat Cell Bio, 2024; Pinkard et al, Nat Comm, 2020), tRNA expression across tissues is highly stable at the isoacceptors level. Our analysis revealed some variations in isodecoder expression in brain vs rest analysis that correlated with ntDRs membership (supplementary figure 6). However, such variations cannot explain the absence of correlation between isodecoder expression and tDRs expression in all other comparisons. Therefore, random tRNA degradation as a source of tDRs cannot explain the observations we have presented, especially the various approaches we used to show statistically significant patterns of tDRs enrichment and source tRNAs.

  1. Section 2.5: It is unclear to me how confident the G-quadruplex formation analysis is. I’m also wondering how enriched are individual tDRs for G-stretches, and how does this enrichment correlate with the pqsfinder scores predicting rG4 formation? Additionally, I’m curious about whether similar rG4 formation is predicted in the parent tRNAs. Is this something only happening in the tDRs or is this an inherited structural feature from parent tRNAs?

Response: First, regarding the confidence, we would say it is moderate. As we wrote in the discussion and in the final paragraph in the results, we refer to these tDRs as potentially G4 quad forming tDRs, not definitive, and more research is required for validation. While we cannot for sure state whether the tDRs we report form rG4s without DMS-seq or SHAPE-seq, tDRs are indeed reported to form rG4s and through these structures they can interact with RNA Binding Proteins (RBPs) (Lyons et al, NAR, 2019; Ivanov et al, PNAS, 2014). There is no evidence that mature tRNAs can fold into rG4 structures. Thus, it appears to be an inherent property of tDRs.

Round 2

Reviewer 1 Report

Comments and Suggestions for Authors

.

Reviewer 2 Report

Comments and Suggestions for Authors

The authors made significant improvements to the study. The manuscript is now acceptable. I would suggest a careful read-through to correct the small number of spelling mistakes.

Reviewer 3 Report

Comments and Suggestions for Authors

Thank you to the authors for addressing all the comments and revising the manuscript. Thank you for clarifications and I appreciate the effort and time invested in making these changes. The revisions have improved the clarity, and quality of the work. It is clear that the authors took the feedback seriously and made thoughtful adjustments to strengthen their study.
Minor: line 27, spelling, capitalization of "Kidney"